# Association of ageing-related biomarkers with peripheral neuropathy in colorectal cancer patients up to 2 years after diagnosis

Wenbo Wu[1,2], Floortje Mols[3,4], Cynthia S. Bonhof[3,4], Lou Maas[1,2], Frederik-Jan van Schooten[1,2], Geja J. Hageman [1,2], Lonneke V. van de Poll-Franse[3,4,5]*, Dounya Schoormans[3,4]*

1 Department of Pharmacology and Toxicology, Maastricht University, Maastricht, The Netherlands, 2 NUTRIM School for Nutrition and Translational Research in Metabolism, Maastricht University, Maastricht, The Netherlands, 3 Department of Medical and Clinical Psychology, CoRPS - Center of Research on Psychological disorders and Somatic diseases, Tilburg University, Tilburg, The Netherlands, 4 Department of Research, Netherlands Comprehensive Cancer Organisation (IKNL), Utrecht, The Netherlands, 5 Division of Psychosocial Research and Epidemiology, The Netherlands Cancer Institute, Amsterdam, The Netherlands

* d.schoormans@tilburguniversity.edu

## Abstract

Peripheral neuropathy (PN) and accelerated biological ageing are common in colorectal cancer (CRC) patients. In vitro and in vivo studies suggest links between biological ageing, oxidative stress, and PN. This longitudinal study examined associations between markers of accelerated ageing (leukocyte telomere length (LTL) and plasma NAD+ levels) and oxidative stress (protein carbonyl content (PCC)) with PN in CRC patients. Newly diagnosed CRC patients (n = 457) were recruited in a Dutch prospective cohort. LTL, plasma NAD+ levels, PCC, and PN (self-reported using the EORTC QLQ-CIPN20) were measured at baseline (prior to treatment), 1-year, and 2-years follow-up. Associations between biomarkers and PN were analyzed using a confounder-adjusted linear mixed model. Longer LTL was associated with higher PN scores, including Sensory PN (SPN) and Motor PN (MPN), while lower plasma NAD+ levels were linked to higher SPN complaints (β:-2.29;95%CI:-4.31,-.27). These associations were primarily driven by inter-individual changes over time. Among chemotherapy-treated patients, lower plasma NAD+ levels were associated with higher total PN scores, SPN, and autonomic PN symptoms. Lower NAD+ levels were longitudinally associated with higher SPN complaints, especially among those treated with chemotherapy. These findings emphasize the potential for targeting NAD+ metabolism to mitigate PN in CRC.

**Data availability statement:** The data supporting the findings of our study are available free of charge for non-commercial research purposes. Approval for the storage and sharing of these data with third parties has been granted by the METC as part of the ethics approval for our research. Researchers can request access via the PROFILES registry (www.profilesregistry.nl), and all reasonable requests for scientific purposes will be honored at no cost. We do not deposit the data publicly due to legal and ethical considerations related to participant privacy, as outlined by the ethics board. However, data can be provided under a data-sharing agreement to ensure compliance with these requirements.

**Funding:** The PROCORE study was funded by the Netherlands Comprehensive Cancer Organisation, Utrecht, the Netherlands; the Center of Research on Psychological disorders and Somatic Diseases (CoRPS), Tilburg University, the Netherlands; and an Investment Grant Large of the Dutch Research Council (2016/04981/ZONMW-91101002). W.Wu is supported by China Scholarship Council (CSC) (grant number 201806160187). The funders had no role in study design, data collection and analysis, decision to publish, or preparation of the manuscript.

**Competing interests:** The authors have declared that no competing interests exist.

## 1. Introduction

Due to the ageing population, early detection, and effective treatment option, the number of colorectal cancer (CRC) survivors is expected to increase [1]. Peripheral neuropathy (PN) is frequently reported by CRC survivors in both short- and long-term and is caused by damaged peripheral nerves. PN can be sensory (SPN) with tingling in feet or hands; motor (MPN), causing muscle weakness; and autonomic (APN) manifesting as lose of balance, poor coordination, and dizziness upon standing [2]. PN can cause difficulties in daily life, negatively affecting health-related quality of life (HRQoL) [3–5]. The pathogenesis of PN however remains largely unknown, whereas this information is crucial for the development of effective prevention and treatment options. Research suggests that accelerated ageing – a discrepancy between biological and chronological age [6] – may be involved, as DNA damage is linked to PN in a mouse model of progeria [7] and age-related changes in inflammation is linked to diabetic PN [8].

Emerging cancer therapies, such as chemotherapy, increase survival rates but also raise the risk of age-related diseases [9–11]. A key mechanism in accelerated ageing is cellular senescence [12], a permanent growth arrest induced by chemotherapeutic agents and radiation. This "treatment-induced senescence" (TIS) can damage normal cells [13,14], and accumulating senescent cells in tissues over time contributes to age-related diseases [13,15].

Telomere length is a well-known marker of cellular senescence and biological ageing [16,17]. Telomeres are repetitive TTAGGG sequences at chromosome ends that shorten due to oxidative stress, particularly due to their high guanine content [18,19]. Protein carbonylation, a common marker for oxidative stress, shows elevated protein carbonyl content (PCC) in nerve damage and neurodegenerative diseases [20,21]. Chronic oxidative stress from chemotherapeutic agents contributes to biological ageing through DNA damage and telomere shortening [22]. Accelerated ageing may further reduce anti-oxidative defenses, making cancer patients more vulnerable to oxidative stress that can induce peripheral nerve damage [23,24].

Ageing coincides with decreased nicotinamide adenine dinucleotide ($NAD^+$) levels, a coenzyme crucial for energy metabolism, DNA repair, and cell signaling [25]. Preserving $NAD^+$ is a promising strategy against ageing-related diseases [26,27], as higher levels improve resilience to oxidative stress and help maintain telomere length and genome stability [27–29].

We hypothesize that accelerated ageing may contribute to the development and progression of PN. This study examines longitudinal associations of ageing markers— LTL, $NAD^+$ (accelerated ageing), and PCC (oxidative stress)—with PN in CRC survivors up to 2 years post-treatment. We also explored modifications by chemotherapy [11], age [26,30], physical activity [31], and tumor type [32,33]. Insights into these associations could guide strategies to prevent and reduce PN in CRC patients.

## 2. Materials and methods

### 2.1 Study design and participants

This study used data from PROCORE, a prospective cohort evaluating the impact of cancer and its treatment on patient-reported outcomes [5]. It included adult patients

from four Dutch hospitals diagnosed with primary CRC between January 1, 2016 and January 31, 2019. Exclusion criteria were prior cancer diagnoses (except basal cell carcinoma), cognitive impairments, or inability to read/write Dutch. Most patients were enrolled shortly after diagnosis, although some (n = 26) previously diagnosed and others already in treatment (n = 16) were included. A detailed description is published elsewhere [5]. The study was approved by the Medical Research Ethics Committees United (approval number: NL51119.060.14), and written informed consent was obtained from all participants.

## 2.2 Data collection and blood processing

Data were collected using the PROFILES (Patient Reported Outcomes Following Initial Treatment and Long Term Evaluation of Survivorship) registry, which is linked to the Netherlands Cancer Registry (NCR) [34]. Questionnaires and blood samples were obtained at diagnosis (n = 457), 1 year (n = 321), and 2 years (n = 264) after diagnosis. Blood was drawn via venipuncture into two BD Vacutainer K2E tubes with EDTA for whole blood and plasma. After centrifugation, 0.5 ml aliquots of plasma and whole blood were stored at −80°C until biomarker measurement.

## 2.3 Sociodemographic, lifestyle and clinical factors

Sociodemographic characteristics (age and sex), and clinical information (cancer stage, primary treatment (chemotherapy, radiotherapy, surgery), and tumor type) were retrieved from the NCR. BMI ($kg/m^2$) was calculated at the time of diagnosis. Educational level, smoking status, physical activity, and comorbidities were self-reported at all time-points [35].

## 2.4 Peripheral neuropathy

PN was assessed with the "European Organization of Research and Treatment of Cancer Quality of Life Questionnaire-CIPN twenty-item scale (EORTC QLQ-CIPN20)", which has three subscales: sensory (SPN, nine items), motor (MPN, seven items), and autonomic neuropathy (APN, two items) [36]. Each item uses a 4-point Likert scale from "not at all" to "very much" and is transformed to a 0–100 scale, with higher scores indicating more severe PN [36]. The overall PN severity was calculated by summing all items.

## 2.5 Measurement of plasma NAD levels and protein carbonyl content (PCC) levels

Thawed plasma samples were centrifuged to use only soluble parts for biomarker measurements. Given the multicenter design and logistical constraints, we measured NAD in plasma rather than in cellular fractions. Plasma total NAD levels ($NAD^+$ and NADH), which have been shown to decline with age and reflect systemic $NAD^+$ status [37,38], were quantified using a sensitive enzymatic cycling assay adapted from Bernofsky & Swan as described previously [39]. To control for potential hemolysis plasma free hemoglobin (Hb) was also measured in triplicate [40,41]. PCC concentration, as marker of oxidative stress, was measured using a 2,4-dinitrophenylhydrazine (DNPH) colorimetric assay at 375 nm [42], in duplicate, and normalized by total protein using the Bradford assay [43]. Absorptions of all colorimetric assays were read on a Model 680XR microplate reader (BioRad, Hercules). All samples of a patient (at diagnosis, one and two years follow-up) were measured in the same run. To control batch effects between different runs, quality control samples drawn from a volunteer were included in each run. The coefficient of variation (CV) within (intra-assay CV) and between different runs (inter-assay CV) of both $NAD^+$ and PCC measurements were below 10% ($NAD^+$ 1.0 and 4.4%; PCC 3.2% and 7.2%).

## 2.6 Measurement of leukocyte telomere length (LTL)

LTL as a marker of biological age, was measured by isolating DNA from whole blood using a QIAamp® DNA Blood Mini Kit (Qiagen), with quality checked by NanoDrop spectrophotometer (Isogen Life Science, Belgium). DNA was stored at −20°C until LTL was measured by monochrome multiplex quantitative PCR (qPCR) [44,45] on a Roche LightCycler 480 machine (Roche) using 384-multiwells plates (Roche, Switserland). Triplicate LTL measures – using 4 ng/µl DNA sample – included

clustering of same-patient samples in a single run, while not using outer wells of the plate, as results deviated from central wells.

DNA of two reference samples with known LTL length (HeLa S3: 5.5 kB, HeLa 229: 14.5 kB) were provided by Prof. Alexander Bürkle, University of Konstanz, Germany. Average PCR efficiencies were 96% for telomere and 94% for β-globulin primers. Telomere length in kB was estimated via regression using HeLa cell lines as references. The Hela S3 (DNA was included in 5 dilutions, 4.0–0.25ng/µL) was used as reference to calculate T/S values and by definition has a T/S value of 1.0, and the average T/S value of the Hela 229 was 3.4 (range 2.6–4.4). Out of 974 samples, 103 (10.5%) were rerun; for 84 (8.6%) CV > 10%, additional replicates resulted in average LTL = 6 [5–9] and for 9 (0.9%) no LTL could be determined. The average LTL was 6.5 ± 0.4(SD) kB. Intraclass correlation coefficients were 0.807(95%CI: 0.783–0.828) for T/S values and 0.823(95%CI: 0.803–0.845) for LTL, indicating good reliability [46]. Nevertheless, all samples from the same individual were measured within the same assay run to minimize between-run variability and limit misclassification bias. Detailed methods and primers are in the Supplementary Information 1 [47].

## 2.7 Statistical analysis

Descriptive analyses compared sociodemographic and clinical characteristics at diagnosis using t-tests/Mann-Whitney for continuous and chi-square for categorical variables. Biomarkers at each time point were described with frequencies, means±SD, or medians (IQR) as appropriate. Pearson's correlation assessed age with accelerated ageing markers LTL and $NAD^+$.

Longitudinal associations between PN scores (total, SPN, MPN, APN scores) and biomarkers (LTL, $NAD^+$, PCC) from diagnosis to 2-year follow-up were evaluated using linear mixed models. Model assumptions (normality of residuals, linearity, and homoscedasticity) were checked and met for all analyses. Follow-up time was treated as a variable (diagnosis, 1-year and 2-year follow-up). LTL data were handled two ways: log-transforming LTL in kB since we introduced two reference samples with known telomere length, and z-score transforming T/S values [48]. $NAD^+$ and PCC were natural-log transformed. A priori selected confounders included age, sex, baseline BMI, chemotherapy, and comorbidities. Plasma hemoglobin was measured to detect erythrocyte lysis during sample processing, given that red blood cells contain high intracellular $NAD^+$ levels and included as confounder in the $NAD^+$ model. As plasma hemoglobin does not reflect systemic hemoglobin concentration or anemia status, it was not used as an indicator of iron deficiency or included in models beyond quality control for $NAD^+$ analysis. As cancer stage and cancer treatment are strongly correlated and including both would risk multicollinearity, we chose to adjust for chemotherapy as a covariate, consistent with our previous study [49]. Random slopes were tested with a likelihood test and included if the model was significant. A hybrid model approach separated intra-individual (within subjects) and inter-individual (between subjects) and biomarker associations using deviation scores and person-mean values across time respectively [50]. Results are presented as β coefficients with 95% confidence intervals (CIs) for total, intra-, and inter-subject effects A sensitivity analysis further adjusted for PN-related co-morbidities, including joint inflammation, liver/kidney diseases, and diabetes [51,52].

Interaction effects of LTL and $NAD^+$ with age, tumor type, vigorous physical activity (MVPA), and chemotherapy were also tested [11,26,30–33]. Significant interactions prompted subgroup analyses by chemotherapy status, age (split at mean), and MVPA (split at median). Prior to analysis, continuous variables were inspected for outliers, which were defined as values greater than ±3 SD from the mean.[53] This conservative criterion is frequently applied to reduce the impact of extreme values on model estimation. Sensitivity analyses were conducted with and without outlier exclusion to evaluate the robustness of our findings. All analyses used Stata 14.0 (MP, StataCorp. 2015. College Station, TX, USA). Given the exploratory and hypothesis-generating nature of this study, no formal corrections for multiple testing were applied. A two-sided p < 0.05 was considered significant.

## 3. Results

### 3.1 Demographics, lifestyle, and clinical characteristics of participants

Of 713 invited patients, 457 (64%) participated at diagnosis. Follow-up was completed by 321 (45%) at 1 year and 264 (37%) at 2 years (Fig 1). Questionnaire return rates were 66.9%, 55.4%, and 37.1%. Previously published details [5,49] show that respondents were younger, more often male, and more likely to have received chemotherapy than non-respondents. At diagnosis, 61.5% of participating CRC patients were male, with a mean age of 67.7 years (SD 8.9), with 31.7% receiving chemotherapy, Table 1.

Compared to patients not receiving chemotherapy (n = 312, 68.3%), those receiving chemotherapy were younger, had higher CRC stage, fewer comorbidities (<2), more frequently received radiotherapy, and underwent surgery less often. There were no significant differences in PN-related comorbidities, including joint inflammation and diabetes.

### 3.2 PN scores and biomarkers concentrations among the course of follow-ups

**3.2.1 PN.** Median PN scores increased at 1-year and 2-year follow-up compared to diagnosis (Fig 2). Among patients who received chemotherapy, total PN and SPN scores were lower at 2-year follow-up than at 1-year, though still higher than at diagnosis. In contrast, MPN and APN scores stayed relatively stable over time regardless of chemotherapy status. Overall PN patterns differed by chemotherapy status, but not significantly.

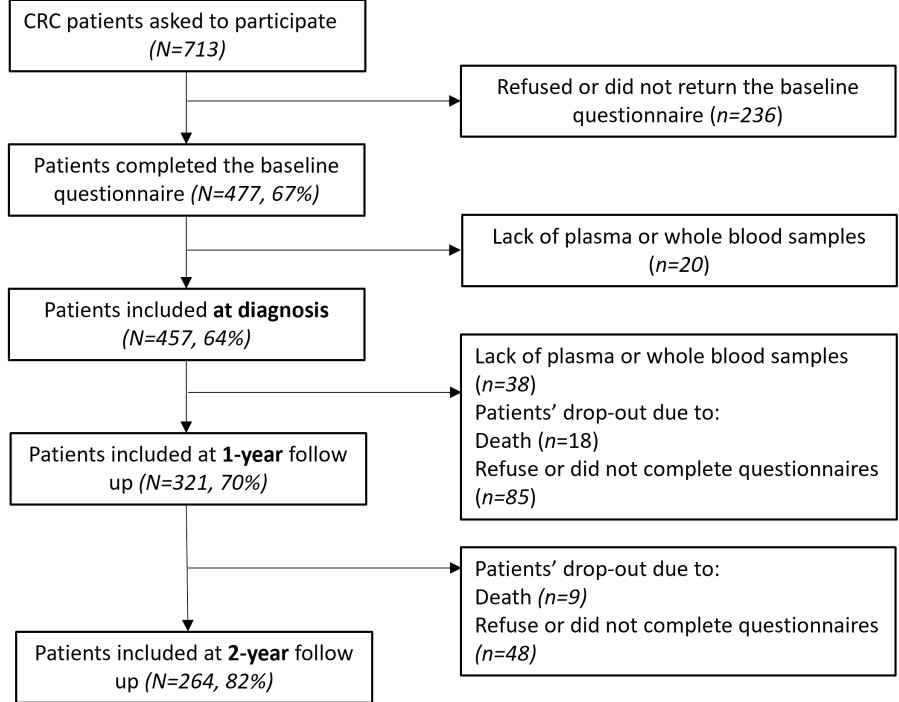

**Fig 1. Flow diagram of this study.** All eligible participants were measured repeatedly at diagnosis, 1-year and 2-year follow ups. Since this current study mainly focused on the biomarkers after colorectal cancer treatment, only available plasma and/or whole blood samples and post-treatment data on the EORTC QLQ-CIPN20 questionnaire, relevant co-variates and self-reported physical activity were included in this study. A detailed description of the PROCORE study design has been published previously [5].

**Table 1. Sociodemographic and clinical characteristics of colorectal cancer patients at baseline (after diagnosis and before starting initial treatment) according to receipt of chemotherapy.**

| | Total (n = 457) | Received chemotherapy (n = 145) | No chemotherapy (n = 312) | p-value |
|---|---|---|---|---|
| Age, mean (SD) | 67.7 (8.9) | 64.0 (8.9) | 68.5 (8.5) | **<0.01** |
| Sex, n (%) | | | | 0.16 |
| Male | 281 (61.5) | 96 (66.2) | 185 (59.3) | |
| Female | 176 (38.5) | 49 (33.8) | 127 (40.7) | |
| BMI, mean (SD) | 26.6 (4.0) | 26.5 (4.1) | 26.6 (4.0) | 0.84 |
| Education Level[a], n (%) | | | | 0.23 |
| Low | 46 (10.2) | 11 (7.6) | 35 (11.4) | |
| Medium | 287 (63.6) | 90 (62.1) | 197 (64.4) | |
| High | 118 (26.2) | 44 (30.3) | 74 (24.2) | |
| Smoking status, n (%) | | | | 0.19 |
| Yes | 53 (11.8) | 21 (14.8) | 32 (10.5) | |
| No | 395 (88.2) | 121 (85.2) | 274 (89.5) | |
| Alcohol consumption, n (%) | | | | 0.44 |
| Yes | 111 (24.8) | 79 (25.9) | 32 (22.5) | |
| No | 336 (75.2) | 226 (74.1) | 110 (77.5) | |
| MVPA (hours/week), Median (IQR) | 11.0 (10.5) | 11.0 (10) | 11.0 (10.5) | 0.052 |
| Complies with Dutch guideline (≥0.5MVPA) | 415 (95.0) | 135 (95.7) | 280 (94.6) | 0.25 |
| Cancer type | | | | 0.66 |
| Colon | 334 (73.1) | 104 (64.1) | 230 (73.7) | |
| Rectum | 123 (26.9) | 41 (35.9) | 82 (16.3) | |
| Cancer Stage | | | | **<0.01** |
| I | 136 (29.8) | 1 (0.7) | 135 (43.3) | |
| II | 132 (28.9) | 11 (7.6) | 121 (38.8) | |
| ≥III | 189 (41.3) | 133 (91.7) | 56 (17.9) | |
| Number of comorbidities, n (%) | | | | **0.01** |
| None | 117 (25.9) | 43 (30.2) | 74 (24.0) | |
| 1 | 157 (34.8) | 58 (40.9) | 99 (32.0) | |
| ≥2 | 177 (39.3) | 41 (28.9) | 136 (44.0) | |
| Comorbidities related to PN[b] | | | | |
| Joint inflammation | 110 (24.1) | 31 (21.4) | 79 (25.3) | 0.36 |
| Liver and/or kidney diseases | 12 (2.6) | 3 (2.1) | 9 (2.9) | 0.61 |
| Diabetes mellitus | 40 (8.8) | 8 (5.5) | 32 (10.3) | 0.1 |
| Received surgery | 444 (97.2) | 133 (91.7) | 311 (99.7) | **<0.01** |
| Received radiotherapy | 78 (17.1) | 47 (32.4) | 31 (9.9) | **<0.01** |

Note: BMI, body mass index; IQR, interquartile range; MVPA, moderate to vigorous physical activities (hours/week); PN, peripheral neuropathy; SD, standard deviation. [a]Education level: Low (no or primary school); medium (lower general secondary education or vocational training); high (pre-university education, high vocational training, university). [b] Most frequent comorbidities associated with peripheral neuropathy. Percentage might not add up to 100% due to rounding off. Statistical significance is denoted in bold.

**3.2.2 Biomarkers.** Biomarker trends were similar across three time points (Supplementary Information Figure S1 and Table S1), despite the presence of some outliers (Fig 3, Supplementary Information Fig S1). At diagnosis, no significant LTL, NAD$^+$, or PCC differences appeared between chemotherapy groups (yes/no) (Supplementary Information Table S2). Age correlated negatively with LTL (rho = −0.219, p < 0.001), but not with NAD$^+$ (rho = −0.02, p = 0.15).

### 3.3 Longitudinal associations of the biomarkers concentrations with PN

**3.3.1 Longitudinal Associations of LTL with PN.** Two different data handling procedures on telomere length were performed, see Table 2 for results. First, a natural logarithm transformation of LTL estimated in kB was included into the linear mixed model. Results showed that log-transformed LTL (in kB) was significantly associated with higher total PN (β: 16.82, 95% CI: 5.60–28.02, p = 0.009), SPN (β: 4.99, 95% CI: 0.54–9.43, p = 0.028), and MPN (β: 6.37, 95% CI: 2.23–10.51, p = 0.003). For total PN, both intra- and inter-individual differences contributed: longer telomere between subjects was linked to a higher total PN score (β: 15.22, 95% CI: 0.46–30.0, p = 0.043), and a within-subject increase associated with higher total PN score (β: 18.79, 95% CI: 3.42–34.15, p = 0.017). SPN increases were driven by intra-individual changes (β: 8.13, 95% CI: 1.57–14.68, p = 0.015), while MPN was explained by inter-individual differences (β: 8.13, 95% CI: 1.57–14.68, p = 0.008). Second, the z-score model of LTL in T/S ratio yielded similar results, this suggests that the two methods of including LTL into the linear mixed models are comparable. Since LTL measured in this study was not following a normal distribution, the linear mixed model can be fitted better by the natural logarithm transformation of LTL (in kB) than using z-scores of T/S ratios.

**3.3.2 Longitudinal associations of NAD⁺ levels with PN.** Lower plasma NAD⁺ levels were inversely associated with higher SPN scores (β: –2.29, 95% CI: –4.31 to –0.27, p = 0.029). Specifically, inter-individual analysis showed that lower average NAD⁺ over time was linked to more SPN (β: –2.69, 95% CI: –5.11 to –0.26, p = 0.030), Table 2. Sensitivity analyses excluding outliers or adjusting for comorbidities did not alter the results (Supplementary Information Table S3).

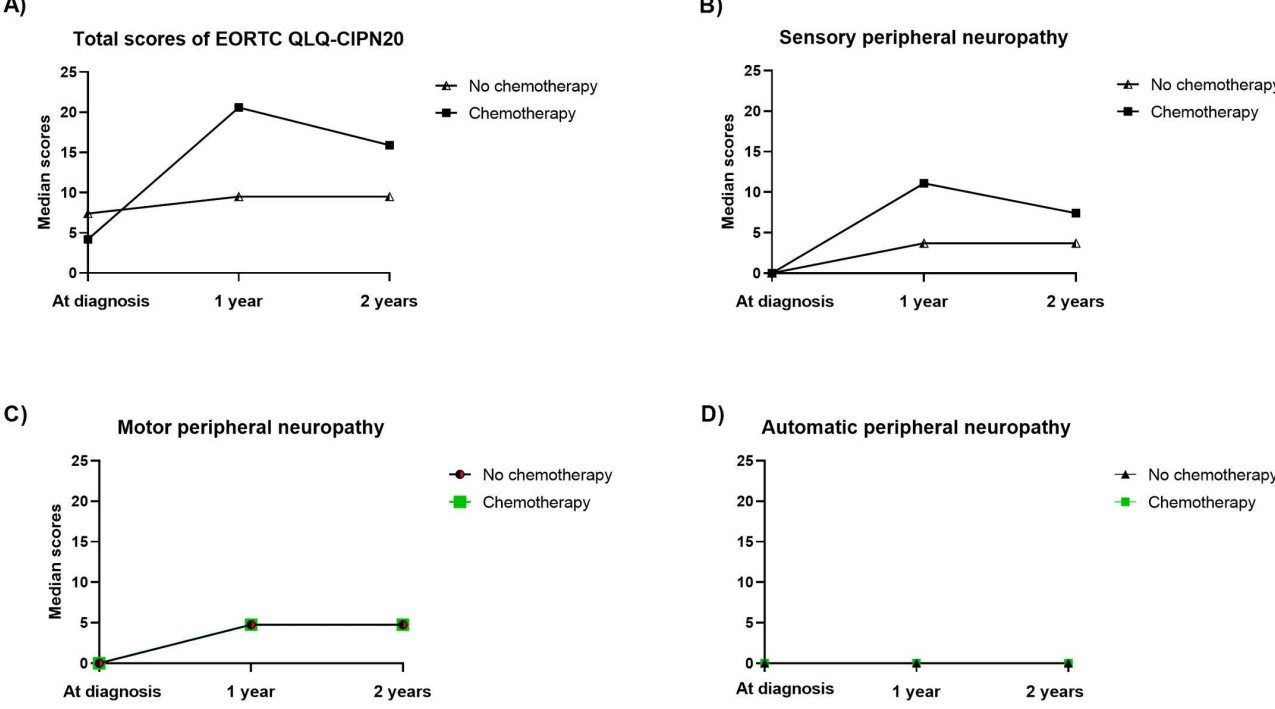

**Fig 2. Median scores of EORTC QLQ-CIPN20 questionnaires (total score, sensory, motor and autonomic subscale scores) of participants who were and were not treated with chemotherapy at diagnosis, and after 1-year and 2-year follow up.** (A) total score; (B) Sensory subscale (SPN); (C) Motor subscale (MPN), the lines of patients who received and did not receive chemotherapy overlapped; (D) Autonomic subscale (APN), the median score was zero for all three time points in both groups. Line types distinguish between chemotherapy and non-chemotherapy groups. Note that overlapping trajectories reflect similar patterns across groups. Y-axis scales were kept consistent to aid comparison.

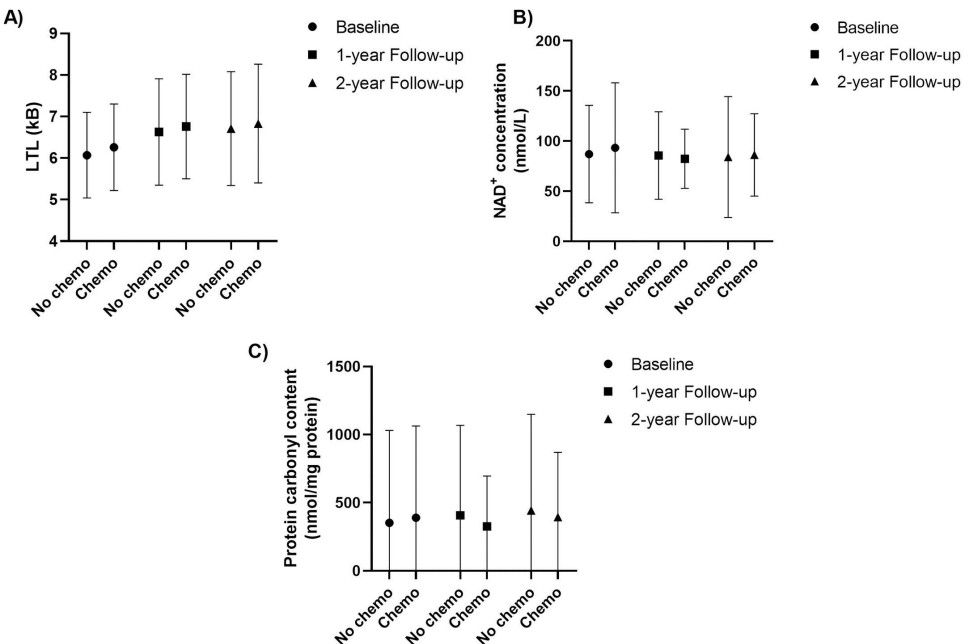

**Fig 3. Measured biomarkers concentrations of participants who were treated with/without chemotherapy at diagnosis, 1-year and 2-year follow up.** (A) Leukocyte telomere length (LTL, in kB); (B) Plasma NAD⁺ levels; (C) Plasma protein carbonyl content (PCC) levels. Each biomarker is plotted using its own y-axis scale to reflect its natural distribution. Lines represent mean values over time. Due to differences in biomarker units, scales are not standardized. Outliers are retained to illustrate the full range of observed values.

**Table 2. Longitudinal associations of LTL, plasma NAD⁺ and PCC levels with PN in colorectal cancer survivors followed-up from the time of diagnosis to 2-year post-diagnosis.**

| | | LTL (in kB)[b] β (95% CI) | LTL in (T/S ratio)[b] β (95% CI) | NAD⁺[c] β (95% CI) | PCC[b] β (95% CI) |
|---|---|---|---|---|---|
| PN total scores[a] | Overall[d] | **16.82 (5.60, 28.02)** | **2.99 (0.82, 5.17)** | 0.37 (−4.80, 5.54) | −1.47 (−4.35, 1.41) |
| | Intra[e] | **18.79 (3.42, 34.15)** | **3.52 (0.32, 6.73)** | 2.50 (−6.28, 11.27) | 3.30 (−5.37, 11.98) |
| | Inter[f] | **15.22 (0.46, 30.0)** | 2.35 (−0.44, 5.13) | 0.41 (−6.15, 6.96) | −2.02 (−5.07, 1.03) |
| SPN[a] | Overall[d] | **4.99 (0.54, 9.43)** | 0.78 (−0.08, 1.64) | **−2.29 (−4.31, −0.27)** | −0.61 (−1.69, 0.46) |
| | Intra[e] | **8.13 (1.57, 14.69)** | 1.37 (−0.05, 2.79) | −0.20 (−3.98, 3.58) | 1.71 (−1.92, 5.34) |
| | Inter[f] | 4.32 (−0.97, 9.61) | 0.43 (−0.58, 1.43) | **−2.69 (−5.11, −0.26)** | −0.82 (−1.95, 0.30) |
| MPN[a] | Overall[d] | **6.37 (2.23, 10.51)** | **1.07 (0.26, 1.88)** | −0.38 (−2.32, 1.55) | −0.29 (−1.39, 0.81) |
| | Intra[e] | 4.79 (−0.89, 10.46) | 0.90 (−0.29, 2.08) | 0.57 (−2.53, 3.66) | 0.79 (−2.30, 3.88) |
| | Inter[f] | **7.52 (1.94, 13.09)** | **1.10 (0.06, 2.15)** | −0.27 (−2.81, 2.27) | −0.43 (−1.61, 0.75) |
| APN[a] | Overall[d] | 4.45 (−0.98, 9.88) | 0.98 (−0.08, 2.05) | 2.74 (−0.00, 5.40) | −0.57 (−2.07, 0.93) |
| | Intra[e] | 5.53 (−1.74, 12.79) | 1.11 (−0.38, 2.61) | 2.20 (−1.97, 6.36) | 1.03 (−3.15, 5.20) |
| | Inter[f] | 2.89 (−4.94, 10.72) | 0.80 (−0.62, 2.22) | 3.07 (−0.49, 6.63) | −0.80 (−2.40, 0.81) |

Note: β, beta-coefficient; CI, confidence interval; PN, peripheral neuropathy; SPN, sensory peripheral neuropathy; MPN, motor peripheral neuropathy; APN, automatic peripheral neuropathy; LTL, leukocyte telomere length; PCC, protein carbonyl content. LTL kB were log transformed; LTL in T/S ratio were z-scores. [a]: Determined by EORTC QLQ-CIPN20 questionnaires. [b]: model adjusted by age, sex, BMI, received chemotherapy (yes or no), and number of comorbidities. [c]: model adjusted by age, sex, BMI, received chemotherapy (yes or no), plasma hemoglobin levels and number of comorbidities. [d]: β (beta-coefficient) indicates the overall longitudinal associations in the outcome score. [e]: β (beta-coefficient) indicates the intra-individual differences in the outcome scores over time within subjects. [f]: β (beta-coefficient) indicates the inter-individual differences in the outcome scores over time between subjects. Statistical significance is denoted in bold.

### 3.3.3 Longitudinal associations of PCC with PN.

There were no overall significant longitudinal associations of PN with plasma PCC levels (Table 2). Sensitivity analyses excluding the outliers or adjusting for comorbidities did change these findings (Supplementary Information Table S3).

## 3.4 Interaction and subgroup analyses

Effect modifiers tested were chemotherapy, age, physical activity (MVPA hours), and tumor type for associations of LTL, NAD$^+$, and PCC with PN scores (Table 3 and Supplementary Information Tables S4-S6). No significant interactions emerged between LTL and chemotherapy. However, NAD$^+$ showed significant interactions with chemotherapy for PN total (p = 0.003), SPN (p = 0.028), and APN (p = 0.006). Stratified analyses revealed overall that in patients who received chemotherapy, higher NAD$^+$ was associated with lower PN total (β: −12.63, 95% CI: −23.74 to −1.52) and SPN scores (β: −5.69, 95% CI: −10.71 to −0.66). These associations were driven by inter-individual differences, PN total (β: −18.83, 95% CI: −32.20, −5.46), SPS (β: −7.84, 95% CI: −13.88, −1.80) and additionally for APN (β: −6.97, 95% CI: −12.92, −1.02). No significant interactions were found for age, MVPA, or tumor type (Supplementary Information Tables S4-S6).

## 4. Discussion

In this longitudinal study among CRC patients, we tracked changes in LTL, NAD$^+$ levels, and PCC over 2 years post-diagnosis and related them to PN. Longer LTL was linked to higher PN total scores, SPN, and MPN, while lower NAD$^+$ levels correlated with increased SPN complaints. These NAD$^+$–SPN associations were mainly due to inter-individual

**Table 3. The overall, intra- and inter-individual longitudinal associations of LTL and plasma NAD$^+$ levels with PN stratified by chemotherapy (yes or no) at diagnosis in colorectal cancer survivors.**

| | LTL (in kB)[a] | | | NAD$^{+}$[c] | | |
|---|---|---|---|---|---|---|
| | Chemotherapy | | | Chemotherapy | | |
| | No β (95% CI) | Yes β (95% CI) | P-interaction[b] | No β (95% CI) | Yes β (95% CI) | P-interaction[b] |
| *PN* | | | | | | |
| Overall[d] | 14.92 (4.47, 25.38) | 23.30 (−3.91, 50.50) | 0.220 | 3.75 (−1.58, 9.07) | **−12.63 (−23.74,-1.52)** | **0.003** |
| Intra[e] | 15.61 (2.87, 28.34) | 70.49 (−17.26,158.24) | 0.476 | 4.13 (−3.70,11.96) | 0.79 (−20.62,22.21) | 0.810 |
| Inter[f] | 9.84 (−7.30, 26.97) | 29.86 (−1.17, 60.89) | 0.091 | 8.26 (0.68, 15.85) | **−18.83 (−32.20,-5.46)** | **<0.001** |
| *SPN* | | | | | | |
| Overall[d] | 4.21 (0.74, 7.68) | 8.84 (−3.06, 20.73) | 0.178 | −0.78 (−2.53, 0,96) | **−5.69 (−10.71,-0.66)** | **0.028** |
| Intra[e] | 6.15 (1.55, 10.74) | 30.35 (−2.24, 62.93) | 0.254 | 0.59 (−1.92,3.11) | −1.14 (−10.99,8.71) | 0.727 |
| Inter[f] | 1.68 (−3.88, 7.24) | **13.09 (0.00, 26.04)** | **0.033** | −0.40 (−2.77,1.96) | **−7.84 (−13.88,-1.80)** | **0.005** |
| *MPN* | | | | | | |
| Overall[d] | 4.47 (0.22, 8.71) | 4.21 (0.74, 7.68) | 0.050 | 0.22 (−1.83,2.26) | −2.62 (−6.59,1.34) | 0.169 |
| Intra[e] | 3.07 (−2.30, 8.44) | 30.35 (−2.24, 62.93) | 0.342 | 1.91 (−1.36,5.18) | −0.69 (−7.92, 6.54) | 0.449 |
| Inter[f] | 4.51 (−1.96, 10.99) | **14.76 (3.60, 25.92)** | **0.028** | 0.89 (−2.16,3.94) | −3.73 (−8.63,1.16) | 0.107 |
| APN | | | | | | |
| Overall[d] | 4.96 (−1.20, 11.11) | 2.95 (−8.02, 13.92) | 0.840 | 5.16 (2.03, 8.30) | **−2.92 (−7.52, 1.68)** | **0.006** |
| Intra[e] | 5.75 (−2.23, 13.73) | 30.35 (−2.24, 62.93) | 0.849 | 1.62 (−3.39,6.63) | 3.42 (−4.10,10.94) | 0.669 |
| Inter[f] | 3.43 (−5.93, 12.79) | 1.91 (−12.48, 16.29) | 0.535 | 7.99 (3.77, 12.21) | **−6.97 (−12.92,-1.02)** | **<0.001** |

Note: β, beta-coefficient; CI, confidence interval; PN, peripheral neuropathy; SPN, sensory peripheral neuropathy; MPN, motor peripheral neuropathy; APN, autonomic peripheral neuropathy; LTL, leukocyte telomere length; a: Models were adjusted by age, sex, BMI, and number of comorbidities. b: Interaction was tested by introducing an interaction term (either "chemotherapy*LTL" or "chemotherapy*NAD$^+$") into the linear mixed modeling. c: Adjusted by age, sex, BMI, plasma hemoglobin levels and number of comorbidities. Statistical significance is denoted in bold. d: β (beta-coefficient) indicates the overall longitudinal associations in the outcome score. e: β (beta-coefficient) indicates the intra-individual differences in the outcome scores over time within subjects. f: β (beta-coefficient) indicates the inter-individual differences in the outcome scores over time between subjects.

changes; lower NAD$^+$ levels were associated with more SPN over time. Subgroup analyses showed lower NAD$^+$ levels were related to higher total PN, SPN, and APN symptoms, especially among chemotherapy recipients.

Over 2 years, PN symptoms persisted and did not return to baseline, aligning with previous findings that PN can persist over time [54]. Chemotherapy recipients reported higher SPN—tingling, numbness, and pain [2,51]—though patients who did not receive chemotherapy also experienced PN, suggesting additional factors like ageing and comorbidities contribute.

Our findings indicate that NAD$^+$ levels are related to PN among CRC patients treated with chemotherapy. Consistently, animal studies show NAD$^+$ metabolism modulates axonal degeneration—a feature of PN [55]. NAD$^+$ fuels enzymes (e.g., PARP1, SIRT1, SARM1) involved in axonal health [26,56]. Inhibiting overactive PARP1 and SARM1 by ABT-888 and DSRM-3716, respectively reduces chemotherapy-induced PN in animal models [57–59]. Lower NAD$^+$ levels were associated with increased SPN in our patients, a finding supported by research using animal models [55,56]. We found significant interactions between plasma NAD$^+$ levels and chemotherapy on PN, suggesting that in CRC patients treated with chemotherapy, lower NAD$^+$ may result in more PN symptoms. Indeed, cancer treatment can accelerate ageing, and the negative link between NAD$^+$ and ageing is well documented [60]. While lower NAD$^+$ levels may result from chemotherapy and/or comorbidities, NAD$^+$-consuming enzymes could also contribute to PN development. While we cannot conclusively state that maintaining NAD$^+$ alleviates or prevents PN, animal studies suggest boosting NAD$^+$ biosynthesis may protect against axonal degeneration [61,62]. Furthermore, recent clinical studies have strengthened the evidence for NAD$^+$ as a promising therapeutic target in humans. Supplementation with NAD$^+$ precursors such as nicotinamide riboside and nicotinamide mononucleotide has shown beneficial effects on cardiovascular health, metabolic function, and inflammatory profiles in older adults.[63,64] This provides human-based support for the translational potential of NAD$^+$-targeted interventions in modulating aging-related processes such as peripheral neuropathy. Our results highlight the potential role of NAD$^+$ maintenance, although future RCTs are needed to test this intervention.

Although there is evidence for a contribution of oxidative stress to the development of PN [65], no associations were found for PCC in our study. In a previous cross-sectional study no associations were found between PCC and QoL, patient reported anxiety and depression in CRC survivors.[39] The absence of an association with PN may be due to a changes in the clearance of plasma protein carbonyls one and two years after diagnosis, that can result in both higher and lower levels than pre-treatment levels, depending on factors such as level of physical activity.[66] Another factor could be the stability of plasma samples during storage. The oldest samples have been stored for more than 6 years before measurement, and although PCC are considered relatively stable, and were reported to be not changed after one month of storage at −70°C [67], it remains to be determined whether plasma PCC is stable after storage for several years.

LTL is recognized as an ageing biomarker [68]. However, despite the evidence gathered from in vitro and in vivo models, epidemiologic studies measuring LTL often report inconsistent results [17,69]. Notably, longer LTL was associated with increased PN symptoms – a counterintuitive finding lacking a clear biological explanation. We measured LTL instead of tissue TL as it is less invasive, but its extrapolation to neuronal tissue is uncertain [70]. LTL is largely dependent on leukocyte turnover, influenced by chronic inflammation, oxidative stress and shifts in specific leukocyte populations [71]. Moreover, telomeres of hematopoietic stem cells may also be damaged during chemotherapy, resulting in an increased shortening of telomeres.[72] Cells with critically shortened telomeres will become apoptotic or senescent, and no longer contribute to the circulating leukocyte population. This results in a selection for stem cells with longer telomeres, and an increase in average LTL in the years following treatment.[73] Therapy-induced damage to non-cancer tissue may result in permanently damaged cells in non-proliferating cells such as neurons, and a selection for cells with longer telomeres in proliferating cells such as hematopoietic stem cells, which may explain the observed association. This raises questions about LTL's validity as a biological ageing indicator, particularly regarding PN. Other ageing biomarkers such as NAD$^+$, AGEs, GDF11/15, DNAm age might offer more precise insights into PN development [26,74–77] reflecting different aspects of ageing.

Study limitations include declining response rates across follow-up waves, which may have introduced attrition bias—particularly if participants with more advanced disease or comorbidities were more likely to drop out due to illness or death—potentially leading to an underestimation of the observed associations between biological aging markers, oxidative stress, and peripheral neuropathy. Due to the multicenter study design, an inclusion period of a little more than three years, and a follow-up of two years, some blood and plasma samples were stored for a period of a little more than six years before analyses were done. Since the stability of PCCs and NAD$^+$ during prolonged storage of plasma at −70°C is not known, this is a limitation of this study. Also the absence of data on neuropathy-related medications such as duloxetine or gabapentin which may influence both the severity and reporting of peripheral neuropathy symptoms is limiting our findings. Additionally, the study did not include information on vitamin deficiencies—particularly B12, B6, B1, and E—which are recognized as important and potentially reversible causes of peripheral neuropathy. Furthermore detailed data on the type and duration of chemotherapy regimens are unavailable, preventing stratified analyses by specific agents such as oxaliplatin, known to be highly neurotoxic. While the EORTC QLQ-CIPN20 is a validated and widely used patient-reported outcome measure for assessing chemotherapy-induced PN, its reliance on subjective reporting may introduce recall or interpretation biases. Future research should address these gaps by including comprehensive data including objective measures of PN symptoms, symptom management strategies, nutritional status, and chemotherapy characteristics to more accurately assess and interpret neuropathy outcomes. Furthermore, due to the potential bidirectional relationships between biomarkers and PN, causality cannot be established; to clarify the directionality of these associations, future intervention studies employing time-lag models are warranted. Despite these issues, the prospective design and repeated measures strengthen our examination of PN's longitudinal links with ageing and oxidative stress markers.

To our knowledge, this is the first study examining PN's longitudinal association with markers of biological ageing (LTL, NAD$^+$ status) and oxidative stress (PCC) in CRC patients up to 2 years post-diagnosis. We found that longer LTL was linked to more PN, and among chemotherapy patients, lower NAD$^+$ was associated with increased PN. As these results suggest NAD$^+$ status is relevant to PN symptoms, future intervention studies should explore maintaining NAD$^+$ levels to prevent or alleviate PN, promoting healthy ageing and improved quality of life post-diagnosis and treatment.

## Supporting information

**S1 Information. Relevant information on reporting leukocyte telomere length.**
(DOCX)

**S1 Fig. Measured biomarkers concentrations of participants at diagnosis, 1-year and 2-year follow up time points. (A) Telomere length (TL, in kB); (B) Plasma NAD$^+$ levels; (C) Plasma protein carbonyl contents levels.**
(DOCX)

**S1 Table. Telomere length, plasma NAD$^+$ levels and protein carbonyl contents levels, and peripheral neuropathy outcomes at post-treatment measurements of participants (data are presented as mean±SD).**
(DOCX)

**S2 Table. Biomarker concentrations of all participants included at diagnosis.**
(DOCX)

**S3 Table. Sensitivity analysis with and without outliers on the overall longitudinal associations of NAD$^+$ and Protein carbonyl content levels with peripheral neuropathy in colorectal cancer survivors followed-up from the time of diagnosis to 2-year post-treatment. Abbreviations: β, beta-coefficient; CI, confidence interval; PN, peripheral neuropathy; SPN, sensory peripheral neuropathy; MPN, motor peripheral neuropathy; APN, autonomic peripheral neuropathy; PCC, protein carbonyl contents.** [a]: Determined by EORTC QLQ-CIPN20 questionnaires. [b]: model adjusted by age, sex, BMI, receive chemotherapy (yes or no), plasma hemoglobin levels

and number of comorbidities. [c]:Model adjusted by age, sex, BMI, receive chemotherapy (yes or no), and number of comorbidities.
(DOCX)

**S4 Table. Subgroup analysis based on tumor type (colon or rectal) of longitudinal associations of NAD+ and telomere length with peripheral neuropathy in colorectal cancer survivors followed-up from the time of diagnosis to 2-year post-treatment. Abbreviations: β, beta-coefficient; CI, confidence interval; PN, peripheral neuropathy; SPN, sensory peripheral neuropathy; MPN, motor peripheral neuropathy; APN, autonomic peripheral neuropathy; TL, telomere length; [a]: Models were adjusted by age, sex, BMI, chemotherapy (yes/no), and number of comorbidities.** [b]: Interaction was tested by introducing an interaction term (either "chemotherapy*TL" or "chemotherapy*NAD+") into the linear mixed modeling. [c]: adjusted by age, sex, BMI, chemotherapy (yes/no), plasma hemoglobin levels and number of comorbidities. Statistical significance was denoted in bold. [d]: β (the beta-coefficient) indicates the overall longitudinal associations in the outcome score. [e]: β (the beta-coefficient) indicates the intra-individual differences in the outcome scores over time within subjects. [f]: β (the beta-coefficient) indicates the inter-individual differences in the outcome scores over time between subjects.
(DOCX)

**S5 Table. Subgroup analysis based on tumor type (colon or rectal) of longitudinal associations of NAD + and telomere length with peripheral neuropathy in colorectal cancer survivors followed-up from the time of diagnosis to 2-year post-treatment.** Abbreviations: β, beta-coefficient; CI, confidence interval; PN, peripheral neuropathy; SPN, sensory peripheral neuropathy; MPN, motor peripheral neuropathy; APN, autonomic peripheral neuropathy; TL, telomere length; [a]: Models were adjusted by age, sex, BMI, chemotherapy (yes/no)and number of comorbidities. [b]: Interaction was tested by introducing an interaction term (either "chemotherapy*TL" or "chemotherapy*NAD+") into the linear mixed modeling. [c]: adjusted by age, sex, BMI, chemotherapy (yes/no), plasma hemoglobin levels and number of comorbidities. Statistical significance was denoted in bold. [d]: β (the beta-coefficient) indicates the overall longitudinal associations in the outcome score. [e]: β (the beta-coefficient) indicates the intra-individual differences in the outcome scores over time within subjects. [f]: β (the beta-coefficient) indicates the inter-individual differences in the outcome scores over time between subjects.
(DOCX)

**S6 Table. The overall, intra- and inter-individual longitudinal associations of telomere length and plasma NAD+ levels with peripheral neuropathy stratified by moderate-to-vigorous intensity physical activity (MVPA) at diagnosis in colorectal cancer survivors.** Abbreviations: β, beta-coefficient; CI, confidence interval; PN, peripheral neuropathy; SPN, sensory peripheral neuropathy; MPN, motor peripheral neuropathy; APN, autonomic peripheral neuropathy; TL, telomere length; [a]: Models were adjusted by age, sex, BMI, chemotherapy (yes/no) and number of comorbidities. [b]: Interaction was tested by introducing an interaction term (either "chemotherapy*TL" or "chemotherapy*NAD+") into the linear mixed modeling. [c]: adjusted by age, sex, BMI, plasma hemoglobin levels, chemotherapy (yes/no) and number of comorbidities. Statistical significance was denoted in bold. [d]: β (the beta-coefficient) indicates the overall longitudinal associations in the outcome score. [e]: β (the beta-coefficient) indicates the intra-individual differences in the outcome scores over time within subjects. [f]: β (the beta-coefficient) indicates the inter-individual differences in the outcome scores over time between subjects.
(DOCX)

## Acknowledgments

We would like to thank all patients and their doctors for their participation in PROCORE. Also, we want to thank the following hospitals for their cooperation: ElisabethTweeSteden Hospital, Tilburg; Catharina Hospital, Eindhoven; Elkerliek Hospital, Helmond; Máxima Medical Centre, Eindhoven and Veldhoven.

## Author contributions

**Conceptualization:** Floortje Mols, Cynthia S. Bonhof, Geja J. Hageman, Lonneke V. van de Poll-Franse, Dounya Schoormans.

**Data curation:** Wenbo Wu, Floortje Mols, Cynthia S. Bonhof, Lou Maas, Dounya Schoormans.

**Formal analysis:** Wenbo Wu, Cynthia S. Bonhof, Lou Maas, Frederik-Jan van Schooten, Geja J. Hageman, Lonneke V. van de Poll-Franse, Dounya Schoormans.

**Funding acquisition:** Floortje Mols, Lonneke V. van de Poll-Franse, Dounya Schoormans.

**Investigation:** Floortje Mols.

**Methodology:** Floortje Mols, Cynthia S. Bonhof, Frederik-Jan van Schooten, Geja J. Hageman, Lonneke V. van de Poll-Franse, Dounya Schoormans.

**Project administration:** Wenbo Wu, Floortje Mols, Cynthia S. Bonhof, Lou Maas, Dounya Schoormans.

**Supervision:** Floortje Mols, Frederik-Jan van Schooten, Geja J. Hageman, Lonneke V. van de Poll-Franse, Dounya Schoormans.

**Validation:** Dounya Schoormans.

**Visualization:** Wenbo Wu, Geja J. Hageman, Dounya Schoormans.

**Writing – original draft:** Wenbo Wu, Floortje Mols, Cynthia S. Bonhof, Lou Maas, Frederik-Jan van Schooten, Geja J. Hageman, Lonneke V. van de Poll-Franse, Dounya Schoormans.

**Writing – review & editing:** Floortje Mols, Cynthia S. Bonhof, Frederik-Jan van Schooten, Geja J. Hageman, Lonneke V. van de Poll-Franse, Dounya Schoormans.

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
