## [Decision Letter · Decision Letter 0]

22 May 2025

PONE-D-25-13325Associating ageing-related biomarkers with peripheral neuropathy in colorectal cancer patients up to 2 years after diagnosisPLOS ONE

Dear Dr. Schoormans,

Thank you for submitting your manuscript to PLOS ONE. After careful consideration, we feel that it has merit but does not fully meet PLOS ONE’s publication criteria as it currently stands. Therefore, we invite you to submit a revised version of the manuscript that addresses the points raised during the review process.

We look forward to receiving your revised manuscript.

Kind regards,

Li Yang, M.D.

Academic Editor

PLOS ONE

Journal Requirements:

 “The PROCORE study was funded by the Netherlands Comprehensive Cancer Organisation, Utrecht, the Netherlands; the Center of Research on Psychological disorders and Somatic Diseases (CoRPS), Tilburg University, the Netherlands; and an Investment Grant Large of the Dutch Research Council (2016/04981/ZONMW-91101002). W.Wu is supported by China Scholarship Council (CSC) (grant number 201806160187).”

4. In the online submission form, you indicated that “The data that support the findings of this study are available from PROFILES registry (www.profilesregistry.nl). Data will be made readily available upon written request sent to the corresponding author with reasonable request, and with possible purchase.”.

Additional Editor Comments:

Thanks for submitting your work to PLOS ONE. Your manuscript has now been assessed by our editorial team and external peer experts. While they found it interesting, you will see that they have raised many serious problems and are advising that you revise your manuscript thoroughly. At the same time, please submit the point-by-point responses to reviewers' comments. If you are prepared to undertake the work required, I would be pleased to reconsider my decision. Please note that this revision decision does not assure the acceptance of your work. Thanks for the opportunity to consider your work.

Reviewers' comments:

Reviewer's Responses to Questions

**Comments to the Author**

1. Is the manuscript technically sound, and do the data support the conclusions?

Reviewer #1: Partly

Reviewer #2: Yes

Reviewer #3: Yes

2. Has the statistical analysis been performed appropriately and rigorously? 

Reviewer #1: Yes

Reviewer #2: I Don't Know

Reviewer #3: Yes

3. Have the authors made all data underlying the findings in their manuscript fully available?

Reviewer #1: Yes

Reviewer #2: Yes

Reviewer #3: Yes

4. Is the manuscript presented in an intelligible fashion and written in standard English?

Reviewer #1: Yes

Reviewer #2: No

Reviewer #3: Yes

5. Review Comments to the Author

Reviewer #1: 1. Scientific Remarks & Literature Alignment

Strengths:

First longitudinal study to explore ageing biomarkers and PN in CRC survivors.

Thoughtful stratification by chemotherapy status and inclusion of intra/inter-individual modeling.

Weaknesses/Points for Clarification:

- Biological paradox: The finding that longer telomeres correlate with higher PN is unexplained. While speculative explanations (e.g., inflammation-mediated leukocyte shifts) are offered, no supporting data are shown. Consider adding CRP/IL-6 as inflammation markers if available.

PCC (oxidative stress): Not associated with PN, despite a strong literature link (e.g., Areti et al. 2014, Redox Biology). Was this due to measurement variability? Report CV% and inter-assay consistency more explicitly.

- NAD+ as a therapeutic target: This is compelling, but would benefit from referencing emerging human studies or clinical trials (e.g., NR supplementation in aging/neurodegeneration). Currently, discussion leans heavily on animal data.

- Confounding by cancer stage or surgery type is not discussed. Could residual confounding explain some of the associations?

- Missing Variables: No mention of neuropathy-related medication (e.g., duloxetine, gabapentin) use, which could bias PN reporting.

2. Language, Grammar, and Style Issues

Below is a list of the most critical issues (by line number or paragraph reference):

Line 1 (Title): Suggest revising to “Association of Ageing-Related Biomarkers with Peripheral Neuropathy in Colorectal Cancer Patients Over a Two-Year Period” for clarity and smoother flow.

Line 50: "autonomically (APN)" → awkward phrasing. Consider "autonomic (APN), manifesting as...".

Line 57: “increase survival” → should be “increase survival rates”.

Line 89: “all participants provided written informed consent.” → better phrased as: “Written informed consent was obtained from all participants.”

Line 144–145: “PN (total, SPN, MPN, APN scores)” → Redundant use of "PN". Suggest: “Peripheral neuropathy scores (total, SPN, MPN, and APN)”.

Line 163–164: Sentence structure is choppy. Consider: “Among 713 invited patients, 457 (64%) participated at diagnosis, with 321 (45%) and 264 (37%) completing follow-up at 1 and 2 years, respectively.”

Line 248: “Surprisingly, longer LTL was associated with more PN symptoms, a finding hard to explain biologically.” → Reword to avoid informal tone: “Notably, longer LTL was associated with increased PN symptoms—a counterintuitive finding lacking a clear biological explanation.”

Line 262: “markers of biological ageing (LTL, NAD⁺ status), and oxidative stress (PCC)” → Comma after “ageing” is unnecessary.

This list is not exhaustive. The manuscript would benefit from thorough copyediting for consistency in punctuation, hyphenation (e.g., “post-treatment” vs. “posttreatment”), and smoother flow in many paragraphs.

3. Statistical and Graphical Review

Statistics:

Appropriateness: The use of linear mixed models (LMMs) is appropriate for longitudinal data. Confounder adjustment is also justified.

Concerns:

Line 185–194 (and Table 2): It’s unclear whether model assumptions (normality, linearity, homoscedasticity) were tested. Please specify model diagnostics.

LTL modeling (log-transformed vs. z-scored) is well justified, but effect sizes (e.g., 35-point change in PN score for a 10% LTL increase) seem clinically implausible. Reassess the scaling of predictors.

Effect modifier analysis (Line 210+): No correction for multiple testing is mentioned. With multiple interaction terms, control for type I error (e.g., using Bonferroni or FDR) is recommended.

Figures:

Figure 2 (PN scores): Clarity issue—chemotherapy vs. non-chemotherapy group lines overlap and aren’t well-distinguished. Use different line types or colors.

Figure 3 (Biomarkers): Axis scaling should be standardized across time points for better comparison. Outliers dominate visualization; consider trimmed mean lines or boxplots with whiskers to mitigate this.

4. Overall Recommendation: Major Revisions

The manuscript addresses a novel and clinically relevant question. The study design is robust, and the statistical approach is broadly appropriate. However, before publication, the following are essential:

- Substantive grammar/spelling corrections.

- Justification of effect sizes and confirmation of model assumptions.

- Clarification and enhanced visualization of figures.

- Deeper biological contextualization of surprising findings (e.g., longer telomeres → worse PN).

- Discussion of potential confounders (e.g., medications, vitamin deficiencies).

Reviewer #2: This study provides valuable longitudinal insights into the associations between ageing-related biomarkers (LTL, NAD+, and PCC) and peripheral neuropathy (PN) in colorectal cancer (CRC) patients. The prospective design, large sample size, and repeated measurements strengthen the findings. The exploration of chemotherapy’s modifying effect on NAD+ and PN is particularly noteworthy, offering potential clinical implications for mitigating PN in CRC survivors. The methodological rigor, including confounder adjustment and sensitivity analyses, enhances the reliability of the results. However, to further strengthen the manuscript, the following revisions are suggested:

1. The association between longer LTL and higher PN scores contradicts existing literature. The authors should discuss potential explanations (e.g., leukocyte turnover, inflammation) and acknowledge limitations of LTL as a proxy for neuronal ageing.

2. Provide details on the enzymatic cycling assay (e.g., sensitivity, specificity) and justify why plasma NAD+ was chosen over cellular NAD+ levels, which may better reflect tissue-specific metabolism.

3. The manuscript mentions outliers but lacks a clear rationale for their exclusion (>±3SD). Include a sensitivity analysis showing results with and without outliers to assess robustness.

4. The EORTC QLQ-CIPN20 is self-reported. Discuss potential biases (e.g., recall, subjectivity) and consider referencing objective PN measures (e.g., nerve conduction studies) for future work.

5. The observational design precludes causal inferences. Explicitly state this and suggest mechanistic studies (e.g., NAD+ supplementation trials) to explore causality.

6. The type/duration of chemotherapy regimens are not detailed. Stratify analyses by specific chemotherapeutics (e.g., oxaliplatin) known to induce PN.

7. Address how declining response rates (64% to 37%) may bias results. Perform attrition analysis comparing baseline characteristics of completers vs. dropouts.

8. The null findings for PCC are underdiscussed. Explore methodological reasons (e.g., assay limitations) or biological explanations (e.g., compensatory mechanisms).

9. Include hemoglobin levels in all models (not just NAD+), as anemia could influence both oxidative stress and PN.

10. The chemotherapy subgroup (n=145) may lack power. Report interaction p-values and confidence intervals to assess reliability.

11. The ICC for LTL (0.807–0.823) suggests moderate reliability. Discuss implications for misclassification bias.

12. Elaborate on how the NAD+-PN association could translate to interventions (e.g., NAD+ boosters). Cite ongoing clinical trials if available.

Reviewer #3: I reviewed this interesting paper examining biomarkers that are usually associated with ageing and their association with Chemotherapy-induced Peripheral neuropathy. The research was conducted in a scientifically sound methodology and there was adequate ethical oversight, which was reported in the manuscript.

The manuscript was easy to read and understand. It described the methodology in a character that was clear and repeatable. The results were presented in full and the write up adequately referenced the results. Of note, the unusual finding of longer Leucocyte Telomere Length (LTL) association with increased CIPN was reported and discussed.

The conclusions were scientific and did not overstate the findings. Suggestions for future work to expand on the findings were made.

I find this study to be neatly described and reported, and I feel it makes an interesting addition to the literature around this important chemotherapy side effect.

Thank you for the opportunity to review the paper.

6. PLOS authors have the option to publish the peer review history of their article (what does this mean? ). If published, this will include your full peer review and any attached files.

**Do you want your identity to be public for this peer review?** For information about this choice, including consent withdrawal, please see our Privacy Policy .

Reviewer #1: No

Reviewer #2: **Yes: ** Mohammad Ebrahimnezhad

Reviewer #3: No

---

## [Editor Report · Decision Letter 1]

14 Jul 2025

PONE-D-25-13325R1Association of ageing-related biomarkers with peripheral neuropathy in colorectal cancer patients up to 2 years after diagnosisPLOS ONE

Dear Dr. Schoormans,

Thank you for submitting your manuscript to PLOS ONE. After careful consideration, we feel that it has merit but does not fully meet PLOS ONE’s publication criteria as it currently stands. Therefore, we invite you to submit a revised version of the manuscript that addresses the points raised during the review process.

We look forward to receiving your revised manuscript.

Kind regards,

Li Yang, M.D.

Academic Editor

PLOS ONE

Journal Requirements:

Additional Editor Comments:

Thanks for submitting your revised paper to PLOS ONE. The previous reviewers are not available this time. Thus, I have carefully assessed your revision work. Based on the manuscript revisions, here are critical questions for the authors to address:

1. NAD⁺ Measurement Validity in Plasma: The manuscript states plasma NAD⁺ was measured due to "multicenter design and logistical constraints," citing Lautrup et al. (2024) to justify its relevance to systemic NAD⁺ status. However, Lautrup’s review primarily discusses intracellular NAD⁺ roles. Please provide additional evidence that plasma NAD⁺ levels reliably reflect tissue-level NAD⁺ bioavailability in neurons, especially given that intracellular NAD⁺ is more directly linked to neuroprotection. Did you validate this against cellular NAD⁺ in a subset of samples?

2. Inconsistency in LTL-PN Association: Longer LTL was unexpectedly associated with worse PN (Table 2). The discussion proposes chemotherapy may select for hematopoietic stem cells with longer telomeres (Benitez-Buelga et al. 2015), but this does not explain why longer LTL correlates with neuropathy severity. Could residual confounding (e.g., unaccounted inflammatory markers or cell-type composition shifts) drive this association? Provide sensitivity analyses adjusting for NLR (neutrophil-lymphocyte ratio) or other immune indices.

3. Data Availability Limitations: Data are accessible via PROFILES registry "upon written request and possible purchase." PLOS ONE mandates unrestricted public access. Clarify how this complies with PLOS’s data policy. Will data be deposited in a public repository (e.g., Dryad) with anonymized identifiers to ensure accessibility without barriers?

4. Chemotherapy Subgroup Heterogeneity: Table 3 shows strong NAD⁺-PN associations in chemotherapy-treated patients but a positive trend (β=3.75, p=NS) in non-chemotherapy patients. Is this divergence biologically plausible? Could unmeasured factors (e.g., specific neurotoxic agents like oxaliplatin) explain this? Provide subgroup analysis by chemotherapy regimen if feasible.

5. Protein Carbonyl (PCC) Stability Concerns: PCC showed no association with PN. The authors note samples were stored >6 years, and stability beyond 1 month at −70°C is unverified (Firuzi et al. 2006). Provide evidence that PCC remained stable under your storage conditions (e.g., correlation between baseline/follow-up levels in controls or spike-in experiments).

6. Attrition Bias in Longitudinal Data: Only 37% completed the 2-year follow-up (page 18), with responders likely healthier than non-responders. How was attrition addressed in mixed models? Provide a comparison of baseline characteristics between completers vs. dropouts to quantify potential bias.

7. Ethical Oversight Clarification: The ethics statement lists approval number "NLS1119.060.14" (page 4), but the Methods section (page 44) cites "NL51119.060.14." Please confirm the correct approval number and ensure consistency throughout the manuscript.

8. I did not see the detailed responses to reviewers' comments, and please provide them just in the cover letter for me to find them.

---

## [Author Response · Author response to Decision Letter 2]

26 Aug 2025

Please read our attached file "r2_response to reviewers" word document for a detailed response and alterations made to the manuscript accordingly.

---

## [Editor Report · Decision Letter 2]

3 Sep 2025

Association of ageing-related biomarkers with peripheral neuropathy in colorectal cancer patients up to 2 years after diagnosis

PONE-D-25-13325R2

Dear Dr. Schoormans,

We’re pleased to inform you that your manuscript has been judged scientifically suitable for publication and will be formally accepted for publication once it meets all outstanding technical requirements.

Kind regards,

Li Yang, M.D.

Academic Editor

PLOS ONE

Additional Editor Comments (optional):

Thanks for the authors' efforts to comprehensively improve your manuscript according to editor's and reviewers' comments. I am pleased to inform you that your paper can be accepted for publication now.
---

## [Editor Report · Acceptance letter]

PONE-D-25-13325R2

PLOS ONE

Dear Dr. Schoormans,

I'm pleased to inform you that your manuscript has been deemed suitable for publication in PLOS ONE. Congratulations! Your manuscript is now being handed over to our production team.

Kind regards,

on behalf of

Dr. Li Yang

Academic Editor

PLOS ONE